# Engineering self-organized criticality in living cells

Blai Vidiella [1,2], Antoni Guillamon [3,4,5], Josep Sardanyés[5], Victor Maull[1,2], Jordi Pla [1,2], Nuria Conde [1,2✉] & Ricard Solé [1,2,6✉]

Complex dynamical fluctuations, from intracellular noise, brain dynamics or computer traffic display bursting dynamics consistent with a critical state between order and disorder. Living close to the critical point has adaptive advantages and it has been conjectured that evolution could select these critical states. Is this the case of living cells? A system can poise itself close to the critical point by means of the so-called self-organized criticality (SOC). In this paper we present an engineered gene network displaying SOC behaviour. This is achieved by exploiting the saturation of the proteolytic degradation machinery in E. coli cells by means of a negative feedback loop that reduces congestion. Our critical motif is built from a two-gene circuit, where SOC can be successfully implemented. The potential implications for both cellular dynamics and behaviour are discussed.

[1] ICREA-Complex Systems Lab, Universitat Pompeu Fabra, Barcelona, Spain. [2] Institut de Biologia Evolutiva (CSIC-UPF), Barcelona, Spain. [3] Departament de Matemàtiques, EPSEB, Universitat Politècnica de Catalunya, Barcelona, Spain. [4] Institut de Matemàtiques de la UPC-BarcelonaTech (IMTech), Universitat Politècnica de Catalunya, Barcelona, Spain. [5] Centre de Recerca Matemàtica, Edifici C, Campus de Bellaterra, Barcelona, Spain. [6] Santa Fe Institute, Santa Fe NM, USA. ✉email: nuria.conde@upf.edu; ricard.sole@upf.edu

In order to adapt to environmental challenges, biological systems exhibit a diverse array of response mechanisms grounded in sensors and actuators as well as in information-processing units. Adaptive responses require dynamical features that combine low energetic costs along with fast changes to efficiently respond to environmental changes. Flocks of birds and fish schools widely fluctuate in time but rapidly reorganize when a perturbation (such as the presence of a predator) occurs. Within cells, the noise was early identified as playing multiple roles affecting cell fate, population heterogeneity, signal amplification or response to stress[1–3]. Noise is both an inevitable outcome of stochastic molecular interactions and an essential ingredient in decision making[4].

It has been shown that many complex systems seem to be poised close to so-called critical points separating ordered from disordered states[5–8]. In a nutshell, both living and non-living systems organize at the boundary separating regular (predictable) from random (disordered) behaviours. At this point, complex dynamics with scale-invariant properties emerge[9,10]. If $s$ defines the total activity in one given event, such as number of firing neurons[11–14], gene expression[15–18], number of active ants in a colony[19,20], critical epidemic bursts[21] or the size of traffic jams[22–24], the resulting distribution $P(s)$ is a fat-tailed one, following a power-law of the form $P(s) \sim s^{-\gamma}$, with a scaling exponent $\gamma$ usually located within the interval $2 \leq \gamma \leq 3$[8,25].

Critical points can be reached by fine-tuning a given control (or bifurcation) parameter. This parameter (such as density of particles, temperature or reaction rate) directly influences the system's state, as described by the order parameter $S$ (system's activity, for example). One way to criticality based on tuning key parameters is well illustrated by enzymatic queueing processes in Ref. [26]. These authors used the framework of queueing theory (QTH) to study the dynamics of different proteins (the customers in QTH) that are processed by a downstream set of enzymes that play the role of servers. They consider the native E. coli protease complexes (ClpXP) which are a limited resource that can only process (degrade) a limited number of incoming proteins. In Fig. 1a we provide a basic diagram considering a protein $\sigma$ being expressed at some given rate $\eta$ (our control parameter). If the rate of protein production is low (queues are short), degradation is efficient since the proteases can process all incoming $\sigma$ units (free phase). If production is too high, a long queue of molecules waiting to be processed will be present (congested phase). The two regimes are separated by a narrow parameter domain (Fig. 1b; see also Supplementary Fig. 2) where an optimal balance is reached, along with broad fluctuations in concentrations[26]. However, they do not follow power laws[27] but exponential-tailed forms $P(s) \sim \exp(-s/s_c)$, see Supplementary Fig. 2b–f, with $s_c$ rapidly increasing as we approach criticality[28]. Here scaling is found to occur instead in the distribution of latencies, i. e. the time required from the production to the final processing of each particle[23].

The critical point is a rather unique one. How can these systems poise themselves into critical states without fine-tuning? An alternative mechanism to reach criticality is provided by self-organized criticality (SOC)[29–31]. In this case, control and order parameters interact in such a way that the system spontaneously self-organizes into a critical state[32,33]. The canonical example of SOC is the critical sandpile (Fig. 1c). By slowly adding grains of sand to the pile (at a rate $\eta$), its slope $\theta$ increases. In the beginning, only a few grains will fall down but the number $s$ of grains in an avalanche rapidly grows as the angle of repose $\theta_c$ is approached. Once we have $\theta = \theta_c$, the interaction between $\theta$ and sand avalanches (the order parameter) will keep the system at criticality[31]. This is summarized in Fig. 1d where the nature of the

feedback between control and order parameters is sketched. However, the concept and its implementation have been controversial and even sandpiles have been found to achieve criticality only under very slow driving and when some microscopic properties are properly tuned[8,34–36]. In this paper, we introduce these minimal conditions of SOC dynamics in living cells by engineering the interaction between order and control parameters in a simple two-gene network design. Given the consensus that the presence of this feedback loop is a pre-condition for SOC, the approach taken here requires locating the SOC motif in the right parameter space (not only a given point) where the scale-free behaviour will be the robust outcome. As shown below, the SOC motif effectively allows driving the system into a bursting dynamical state where gene expression levels (our queue size distributions) follow power laws.

## Results

**Two-gene SOC motif model: deterministic and stochastic dynamics.** The importance of the queueing dynamics in the enzymatic processing is illustrated by the E. coli stress response to starvation, which is triggered by an excess of mistranslated proteins. Stress can cause a significant increase in the concentration of aberrant proteins, which must be degraded. When such an overload occurs, the concentration of the sigma factor (the master stress regulator) builds up, eventually triggering the stress response[37]. Recent theoretical work also suggests that queueing could be adaptive in parallel enzymatic networks when the input flux of substrates is balanced by the maximum processing capacity of the network[38]. Here we go a step further and show how a simple SOC circuit can be actually engineered in vivo.

Our goal in this work is to define the basic design principle to build a genetic sandpile system that captures the feedback structure shown in Fig. 1e and generates bursting, fat-tailed avalanches of activity. First of all, consider the simple, two-gene network circuit shown in Fig. 1e. Two proteins $\sigma_1$ and $\sigma_2$ resulting from their expression will be used as the building blocks for the order and control parameters, respectively, thus implementing a SOC feedback loop (Fig. 1f). The aim is to exploit the topology of the gene-gene interaction in such a way that the system can detect the degree of congestion of the ClpXP system by using $\sigma_2$ as a sensor of the $\sigma_1$ levels. Our control protein $\sigma_2$ can form dimers, i. e. $\sigma_2 + \sigma_2 \rightarrow \sigma_2\sigma_2$ and these dimeric forms act as inhibitors (see Methods, Eqs. (1–3)). If congestion occurs, the abundance of $\sigma_2$ increases and its negative feedback effects also do so. A standard Hill function will be used to model the dimers as transcriptional repressors. The construction of our circuit involves two steps: (a) engineering the critical motif and (b) adjusting the protein production levels. This might seem contradictory with the self-organized description of SOC, but an example of why this is required is given by rice piles[34] which exhibit SOC for some given grain aspect ratios.

Along with the topology of the SOC motif, separation of scales is known to be a characteristic of SOC[33]. While the control parameter has slow dynamics (the angle of the sandpile) the system's response (the avalanche time scale) is fast. In our model, two additional parameters are used to favour the presence of criticality. These are the promoter efficiency for $\sigma_2$, labelled $\eta_2$; and an extra inhibition acting on the repressor $\sigma_2\sigma_2$ indicated as $\mu$ in Fig. 1e. We need to remember that degradation (and other dissipative events) affects $\sigma_2$ and thus a minimal concentration of this sensor is needed in order to effectively detect congested states. On the other hand, in order to experimentally validate our model, we need to tune the strength of the feedback (required to trigger a rapid decay of the intracellular concentration of $\sigma_1$). By

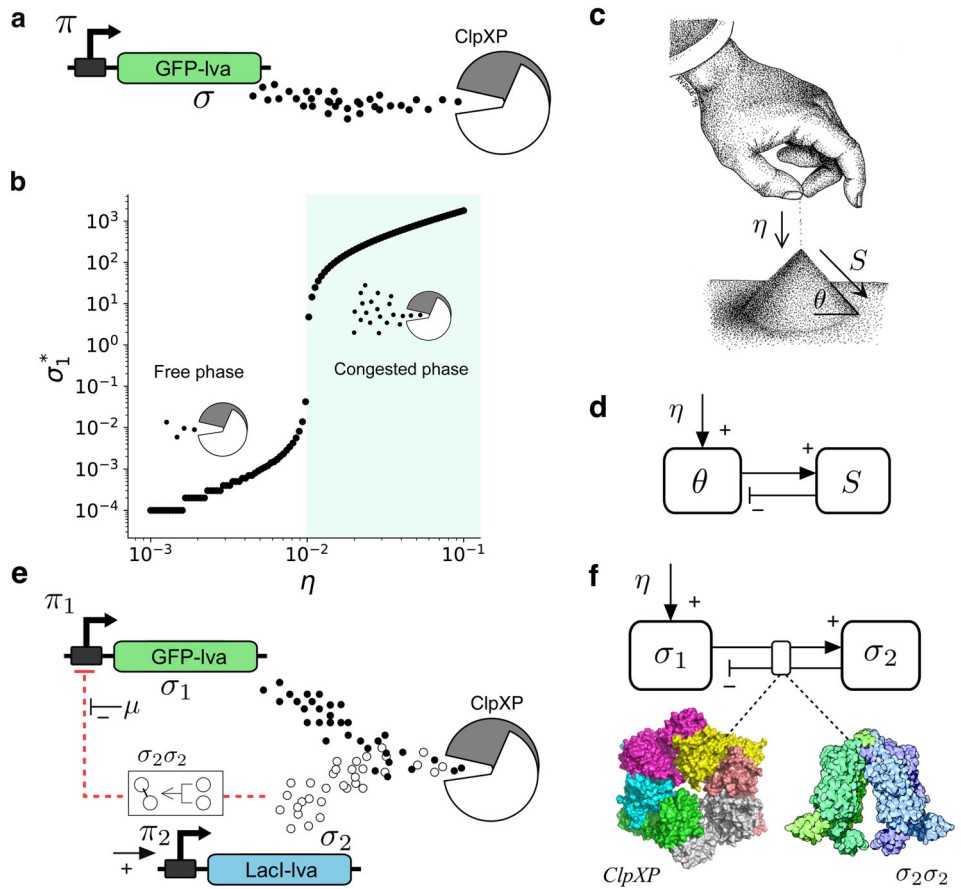

**Fig. 1 Paths to intracellular criticality.** Tunable critical dynamics can be found in simple genetic circuits (**a**) where a given gene (here coding for GFP-Iva) is constitutively expressed into a protein $\sigma$ that decays and is also actively degraded by the cell proteolytic machinery (ClpXP). By tuning expression rate $\eta$ (**d**), a critical rate $\eta_c$ is found to separate a phase of efficient degradation from another involving congestion. In (**b**) the thick line indicates that few proteins are found for $\eta < \eta_c$ (the proteolytic machinery efficiently degrades it) while it accumulates on the right side, due to congestion (light green; ClpXP fails to degrade all the incoming proteins). An alternative, non-externally tuned path is self-organised criticality (SOC), provided by the sandpile (**c**, adapted from @ricard_sole Oct. 2017). As grains of sand are slowly added at a rate $\eta$, the angle of the pile $\theta$ grows and only small avalanches will be observed. However, as the critical (maximum) $\theta_c$ is reached, avalanches of all sizes take place, reducing $\theta$. The feedback between the order parameter ($S$) and the control parameter ($\theta$) is summarised in (**d**). To facilitate the conditions enabling SOC, a two-gene circuit with negative feedback (**e**) allows mapping the sandpile feedback diagram (**f**). Here, both proteins compete for ClpXP (higher levels of $\sigma_1$ also implies high values of $\sigma_2$) and repression feedback is mediated by $\sigma_2\sigma_2$ (the Lac repressor dimer) with $\sigma_1$ and $\sigma_2$ acting as order and control parameters, respectively. Protein models generated using Pymol Software.

tuning these two parameters, we include both the non-SOC design based on queueing as a special case[26] (see Section I in the SM) and a mechanism to achieve the SOC state.

In Fig. 2 we summarize the behaviour of the two-gene system (Fig. 1e; see Methods and Section II in the SM for mathematical details) using both deterministic (Section II.A, SM) and stochastic (Section II.B, SM) dynamics. A unique stable equilibrium ($\sigma_{eq} = (\sigma_{1,eq}, \sigma_{2,eq})$, indicated with a solid black circle in the ($\sigma_1, \sigma_2$) phase portraits of Fig. 2) is found, with a characteristic structure of the orbits in the phase space, as shown in Fig. 2a for the unregulated domain (here $\eta_2 = 10^{-3}$). As the expression rate $\eta_2$ of the control parameter increases close to $\eta_2 \sim 10^{-2}$, it is easy to see the presence of slow-fast dynamics in the distinct structure of the vector fields consistently with the SOC requirement of time scale separation. Here the critical motif allows for large fluctuations in $\sigma_1$ to occur (Fig. 2b) as shown by the compression of the trajectories in the phase portrait close to the fixed point, to be compared with the more homogeneous flow displayed in Fig. 2a for $\eta = 10^{-3}$. The analysis of this system shows that, once close to the equilibrium point, small changes in the control $\sigma_2$ trigger marked population spikes in $\sigma_1$ (see Supplementary figure 4).

Larger values for $\eta_2$ (Fig. 2c) do not exhibit such a time scales separation. The analytic and numerical investigation of the eigenvalues of the fixed point to study its stability properties reveal the presence of a maximum in the ratio of imaginary to real parts (See Supplementary Fig. 7), indicating a remarkable change in the vector field of the phase portrait when $\eta_2 \sim 10^{-2}$ (given our set of fixed parameters indicated in Fig. 2).

To see how these nonlinear flows behave under the presence of intrinsic noise, a stochastic numerical implementation of the two-gene circuit has been carried out using the Gillespie method[39]. In Fig. 2d the coefficient of variation $CV = \sqrt{\langle\sigma_1^2\rangle - \langle\sigma_1\rangle^2}/\langle\sigma_1\rangle$ of the generated time series is displayed against $\eta_2$ for three values of $\mu$. This coefficient provides a statistical estimate of the variance of the fluctuations and a well-defined maximum is observed when $\eta_2 \sim 10^{-2}$. In Fig. 2e we have overlapped several stochastic realisations with the vector field close to the maximum $CV$ (for $\eta_2 = 10^{-2}$). The density plot reveals that the stochastic system visits very frequently the fixed point $\sigma_{eq}$ (orange-red colours in Fig. 2e), but also wanders far away in the lower part of the phase portrait, where the vector field is faster and pushes the stochastic paths far away from the equilibrium point, then returning it back

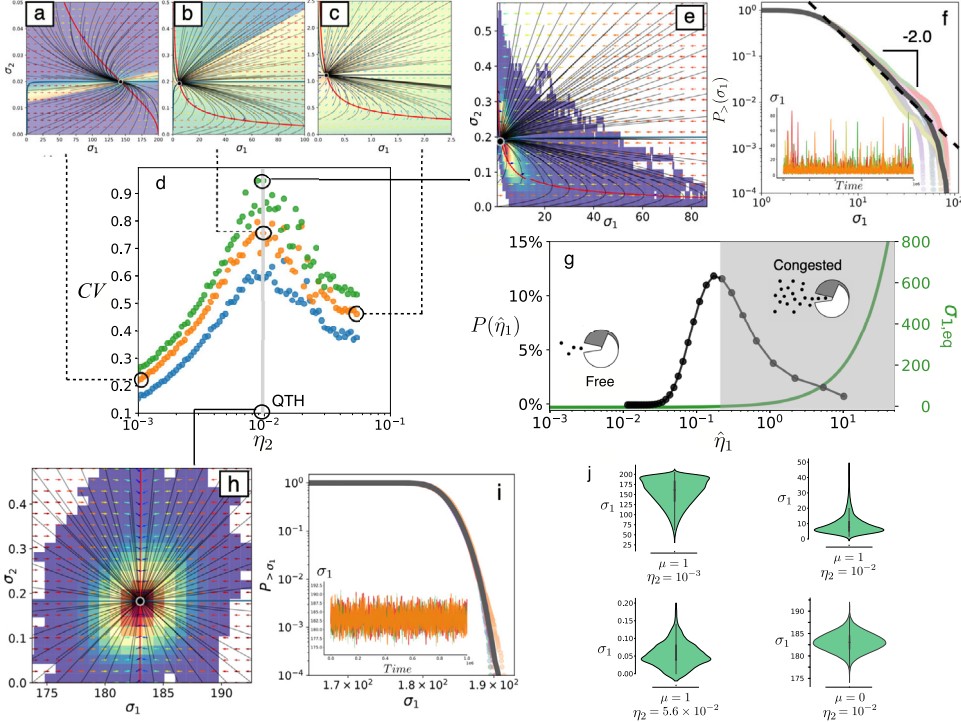

**Fig. 2 Nonlinear dynamics of the two-gene critical motif.** (**a–c**) Orbits for Eqs. (1–2) in the ($\sigma_1$, $\sigma_2$) space with $\eta_2 = 10^{-3}$ (**a**), $\eta_2 = 10^{-2}$ (**b**), and $\eta_2 = 0.056$ (**c**), setting $\eta_1 = 10^{-2}$, $\delta_{1,2} = 5 \times 10^{-2}$, $\delta_c C = 10^{-2}$, $K = \theta = 10^{-3}$. The nullclines are plotted in red ($d\sigma_1/dt = 0$) and blue ($d\sigma_2/dt = 0$). The colour of the arrows of the vector field corresponds to their module (blue: small; red: large). The background colour shows the arrival times to the attractor (shorter times in yellow; longer times in violet). The stochastic dynamics of the model reveals a maximum in the CV when $\eta_2 \sim 10^{-2}$, as shown in panel (**d**) where the colours stand for $\mu = 0.5$ (blue), $\mu = 1.0$ (orange) and $\mu = 1.5$ (green). The relative location of the deterministic flows is indicated by dashed lines. Three different values of the coupling parameter $\mu$ are used to show the robust nature of the maximum of CV, where the SOC motif has been tuned to generate fat-tailed behaviour, as shown in panel (**e**), where the hot map is overlapped to the phase space, showing a larger density close to the deterministic attractor as well as the fat-tailed scaling behaviour (**f**) with $P_>(\sigma_1) \sim \sigma_1^{-2}$ which gives $\gamma \approx 3$ for $P(\sigma_1)$. Here five distributions are shown for 5 independent runs along with their average (dark line). (**g**) Transition between free and congested phases (green curve, in linear-log scale), computed using the same parameter values as in (**e**) and (**f**), and located at $\eta_1 \approx 0.02$. The solid black line with dots shows how the value of the $\sigma_1$ inhibition function, here labelled as $\hat{\eta}(t)$, is tuned by the system itself driving it close to the transition value (see also Supplementary Figures 9–11). Here $P(\eta_1)$ is the probability of $\hat{\eta}(t)$ to take different values of $\eta_1$. By contrast, the flows and hot maps for the non-SOC circuit close to the queuing theory transition (**h**) have a Gaussian pattern (**i**) with exponential tails as shown by the straight lines in the linear-log insets (here $\eta_2 = 10^{-2}$ and $\mu = 0$). The distributions are depicted with the violin plots in panel (**j**) for the indicated parameter values.

to the deterministic equilibrium. In Fig. 2f the resulting distribution is displayed. Specifically, if $P(\sigma_1)$ indicates the probability distribution of $\sigma_1$ expression levels (activity), the cumulative distribution is defined as $P_>(\sigma_1) = \int_0^{\sigma_1} P(\sigma) \, dP(\sigma)$ and helps smoothing the random noise exhibited by $P(\sigma_1)$. If the original distribution follows a scaling $P(\sigma_1) \sim \sigma_1^{-\gamma}$, the cumulative one gives $P_>(\sigma_1) \sim \sigma_1^{-\gamma+1}$. The stochastic model gives a value of $\gamma \sim 3$ (Fig. 2f) estimated from the average of five different runs. The time series associated with this parameter combination is shown in the inset, revealing a characteristic bursting dynamics typical of the SOC state, to be compared with the smooth, single-scale behaviour of the unregulated dynamics setting $\mu = 0$ (Fig. 2h–i) (see also Supplementary Figure 2).

The distinct nature of the SOC motif action is captured by looking at the dynamics of the effective driving term (Eq. (3) in Methods) that we label as $\hat{\eta}_1$ (Eq. (S.17), SM), as summarized in Fig. 2g (see also SM). We can appreciate how this dynamic driving behaves by computing the probability (labeled as $P(\hat{\eta}_1)$) of finding the system at some given driving value $\hat{\eta}_1$. This distribution is peaked close to the free-congested transition (see also Supplementary Figures 10–11 where parameters $\eta_{1,2}$ are tuned to check the robustness of the identified SOC behaviour to these parameters). However, there are no true phases now: the

system bounces back and forth between fluid and congested states as it tends to get close to criticality (see also Section II.A.3, SM). By exploiting the SOC motif, which allows reducing the rate of $\sigma_1$ production, the Poissonian dynamics of the original, non-regulated queueing dynamics is transformed into a bursting signal with fat-tailed statistics.

**Engineering a synthetic SOC circuit in *E. coli*.** The theoretical and computational modelling predicts that the SOC feedback loop defined above (Fig. 1e–f) will display bursting dynamics with fat-tailed activity distributions associated with the $\sigma_1$ protein (order parameter). If the expression level ($\eta_2$) of the control protein $\sigma_2$ is large enough, its concentration will act as a congestion sensor and will repress $\sigma_1$, following the SOC motif design. Otherwise, the system will lack the feedback loop. Here we show how can we engineer the genes circuit incorporating some parameters that allow including the non-SOC phase transition as described above including intracellular queueing processes as a special case. Since the time scale of expression changes can not be captured at the single-cell level because we may need long time series and cells replicate in short times, no individual time series can be gathered. Instead, the collective response of the system will be analyzed to detect the presence of a SOC state. The underlying

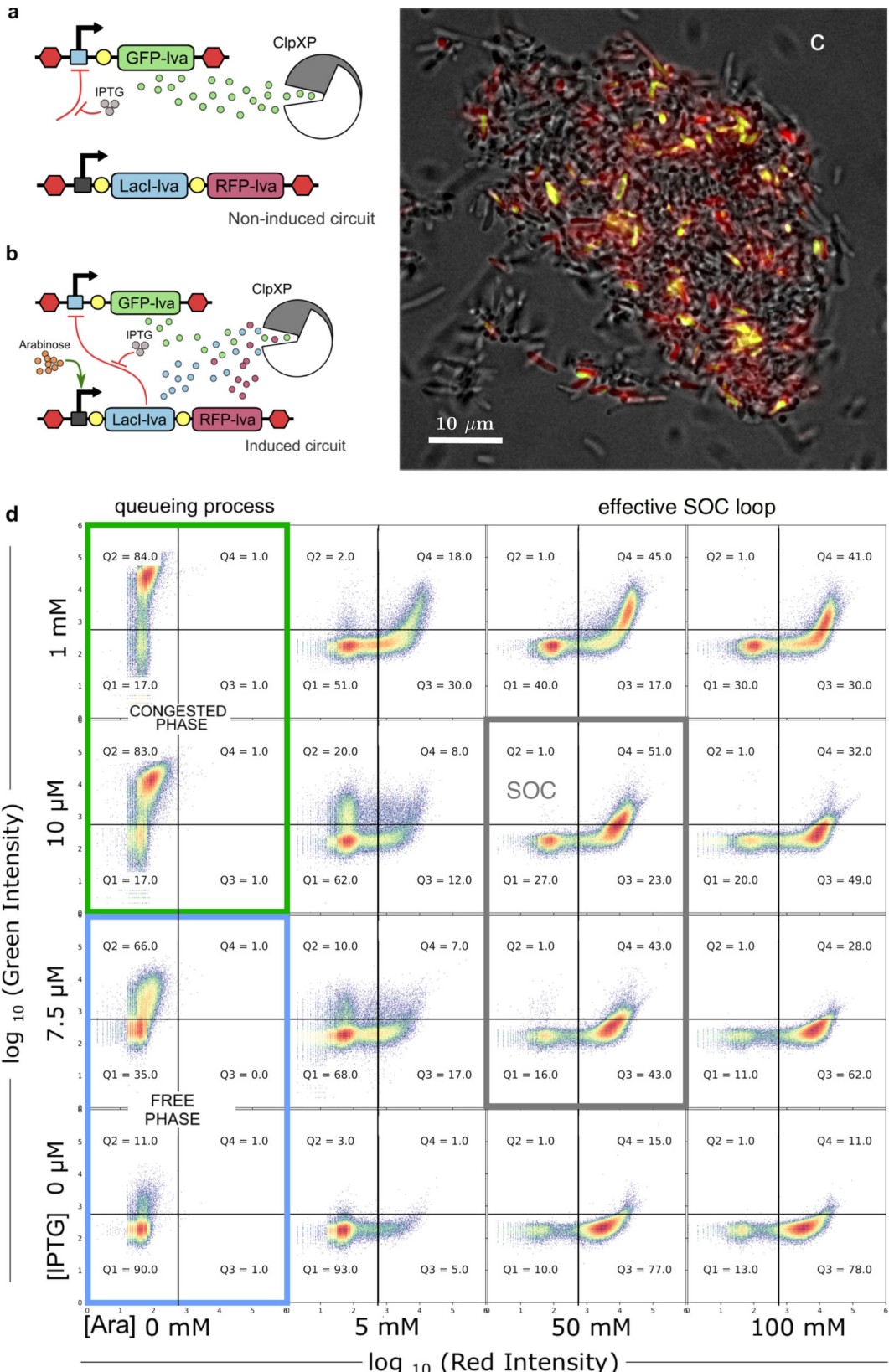

assumption is thus that we have a colony-level sampling of the dynamical states and the critical dynamics is assessed by looking at the distribution of cell states and the resulting aggregated statistics (assuming ergodicity).

The explicit experimental design of our SOC circuit implementation is outlined in Fig. 3a–b. The order parameter is encoded by the green fluorescent protein (GFP), and the control parameter acting as the congestion sensor, by the *LacI* repressor protein, the relative expression of which will be estimated by means of the expression of the red fluorescent protein (RFP). In both cases, we use the unstable variants GFP-lva and RFP-lva, respectively. The construct expresses GFP under the *pLacI*

**Fig. 3 Engineered gene circuit implementing the SOC motif in *E. coli*.** Gene constructs considering non-induced (**a**) and induced (**b**) states. (**c**) Overlapped bright field and fluorescence images of bacteria-induced with Ara (100 mM) and IPTG (10 $\mu$M). Yellow bacteria express both GFP and RFP. **d** Flow cytometry dot plots (green vs red channel) of *E. coli* cultures exposed to different concentrations of IPTG and Ara. In the non-induced circuit (**a**), without Ara, the transition from noncongested proteolytic machinery phase (i.e. free phase, subcritical) to congested phase (supercritical) depends on the tunable GFP-lva production. As IPTG increases, so do the de-repression of GFP expression, ClpXP is not able to degrade the excess of GFP and most cells exhibit high green emission. When the circuit is induced (**b**), Ara triggers the expression of the LacI repressor and the RFP reporter, that are also degraded by the ClpXP complex. The increase of tagged proteins to be degraded contributes to the congestion of ClpXP but also the *LacI* repression helps to de-congest by reducing the tagged GFP expression. Hence, as Ara concentration is increased, a shift towards higher RFP levels along with dispersal and lower levels of GFP values is observed. This defines the parameter space domain (grey window) where the feedback loop required for SOC is at work. For the larger concentrations of IPTG and Ara, the SOC loop remains effective, at least within the limits of the experimentally explored parameter space. A SOC state is obtained in the presence of high Ara concentration around the IPTG values close to the queueing transition. Most cells (Q3 + Q4 ≈ 80%) are emitting in the red channel, but exhibit a broad range of green fluorescence levels, since this state is characterized by fluctuations associated with large bursts of GFP expression (the heterogeneous GFP expression is apparent in the yellow cells of (**c**) and in the histogram of Fig. 4). Microscope images of this experiment and FACS of more IPTG-Ara combinations are shown in Section III.D of SM and in the source data file.

promoter ($\eta_1$) while the *LacI* repressor and RFP reporter protein are under the *pBAD* promoter, with nonleaky tight regulation and high-level expression inducible by Arabinose ($\eta_2$)[40–42]. All three proteins of the circuit are tagged with lva sequence to be degradable by the ClpXP proteolytic complex.

ClpXP is responsible for degrading proteins carrying the SsrA or YbaQ degron sequences, reducing the half-life of a tagged protein from hours to minutes. In an exponentially growing *E. coli* culture (Optical Density (OD) $OD_{660}$ from 0–2), the endogenous levels of ClpX and ClpP are constant and involve around 100 ClpXP molecules, which can degrade at least $10^5$ molecules of GFPssrA per cell per replication cycle. However, due to the limited number of ClpXP protease complexes, the degrading capacity of ssrA-tagged proteins can be easily saturated by the overproduction of a synthetic tagged protein[43–45].

The non-induced circuit depicted in Fig. 3a is thus a particular instance of our more complex motif (Fig. 3b) that would correspond in our case to the presence of endogenous LacI, with the GFP-lva expression being repressed. The presence of Arabinose leads to a strong expression of the repressor LacI-lva (our control parameter) and the reporter protein RFP-lva. Only when high levels of the -lva tagged proteins are reached, and the degradation machinery is saturated, there is enough LacI-lva to repress the production of GFP-lva. The repression loop is consistently removed when the production of GFP-lva (our order parameter) is reduced enough as for desaturate the ClpXP protease that can degrade the repressor again. The addition of Isopropyl $\beta$-d-1-thiogalactopyranoside (IPTG), as indicated by a negative input to the repression feedback (see Fig. 3b), switches on our SOC circuit and allows to control the level of GFP-lva expression. High levels of IPTG will lead to an overproduction of GFP-lva and the subsequent ClpXP complex congestion, thus reproducing the limit case that would correspond to the standard phase transition of the queueing process[26] (Fig. 1a–b).

The SOC motif is contained in a single, high-copy plasmid to ensure a maximal concentration of the vector while maintaining the parity of the two parts of the circuit (Fig. 3a–b). The construct was transformed in the XL1-Blue *E. coli* strain. Further details of the cloning process and sequences can be found in Section III, SM. Also, this design allows easy tuning to obtain parameters involving SOC, in terms of the strength of the promoter (i. e. pBAD) and by adjusting the efficiency of the repressor (in our case, IPTG for LacI, see Section III, SM). To perform the experiments, a single colony was inoculated in a volume of 4 ml, and grown at 37 °C until the exponential phase was reached with an $OD_{660}$ around 0.6. This homogeneous fresh culture was then used to inoculate all the conditions used in the experimental design. Each combination was inoculated with 1 $\mu$l of the starter culture to a final volume of 4 ml. Cells were grown for about 10 h at 37 °C, until reaching an approximate $OD_{660}$ of 0.8–1. The

output of each condition was then analysed using both Fluorescence-Activated Cell Sorting (FACS) and fluorescence microscopy (Fig. 3c–d).

The results from the FACS are displayed in Fig. 3d, where a $4 \times 4$ array of different combinations of IPTG and Arabinose concentrations define our parameter space by means of dot plots. The range of concentrations shown here is $0 \le$ mM [IPTG] $\le 1$, $0 \le$ mM [Ara] $\le 100$. As described above, these small molecules allow to explore a parameter space where we can move from a decoupling between the two genes to full-fledged repression feedback required for criticality to occur. The different cell population responses to the tuning of both IPTG and Ara reveal the relative impact of each on the SOC motif. The target for a SOC state implies two requirements: (i) the expression of large enough levels of the control parameter to effectively perform its feedback; and (ii) a GFP expression characterised by bursts but displaying a low average activity. In the non-induced state, without arabinose (left column), increasing levels of IPTG concentration promote a standard transition from the free to the congested phase (bottom and at the two top panels, respectively). As IPTG grows, we effectively weaken the strength of the repression loop until a critical point is reached allowing congestion to rise. This is observed from the displacement of the density dot plots from low to high levels of GFP. Notice that here distributions appear peaked, as expected from the single-scale theoretical prediction for queueing processes.

In order to generate bursting patterns, Arabinose concentration needs to be increased. To achieve this, we move in the other dimension (horizontal direction in Fig. 3d) of our parameter space, where the control molecule gets more common (cells emit in the red channel) but cannot always effectively act as a repressor. This clearly is a time point picture shot of a bacterial population that exhibits the fluctuating GFP levels characteristics of the SOC state. We emphasize that the distributions in the SOC zone (grey rectangle in Fig. 3d) are well delineated in the parameter space explored experimentally. Outside this region, the SOC loop for larger IPTG and Ara concentrations seems to be still effective (bursting dynamics is expected to occur here too). An example of SOC behavior is shown in the fluorescence microscope image of Fig. 3c, where some bacteria do not have the ClpXP saturated (and thus do not display fluorescence), many are near the critical state of ClpXP saturation with lower levels of effective LacI-lva to repress the GFP, and exhibiting a wide range of GFP-lva concentrations (bacteria in yellow) and few bacteria have enough LacI-lva to degrade the GFP (bright only in red).

The experiments with *E. coli* confirm SOC fluctuations of GFP-lva, while the reporter of the control element (i. e. RFP-lva) remains basically stable in terms of the concentration levels and their dispersal. The experimental system successfully reproduces another important feature of criticality, namely the presence of a

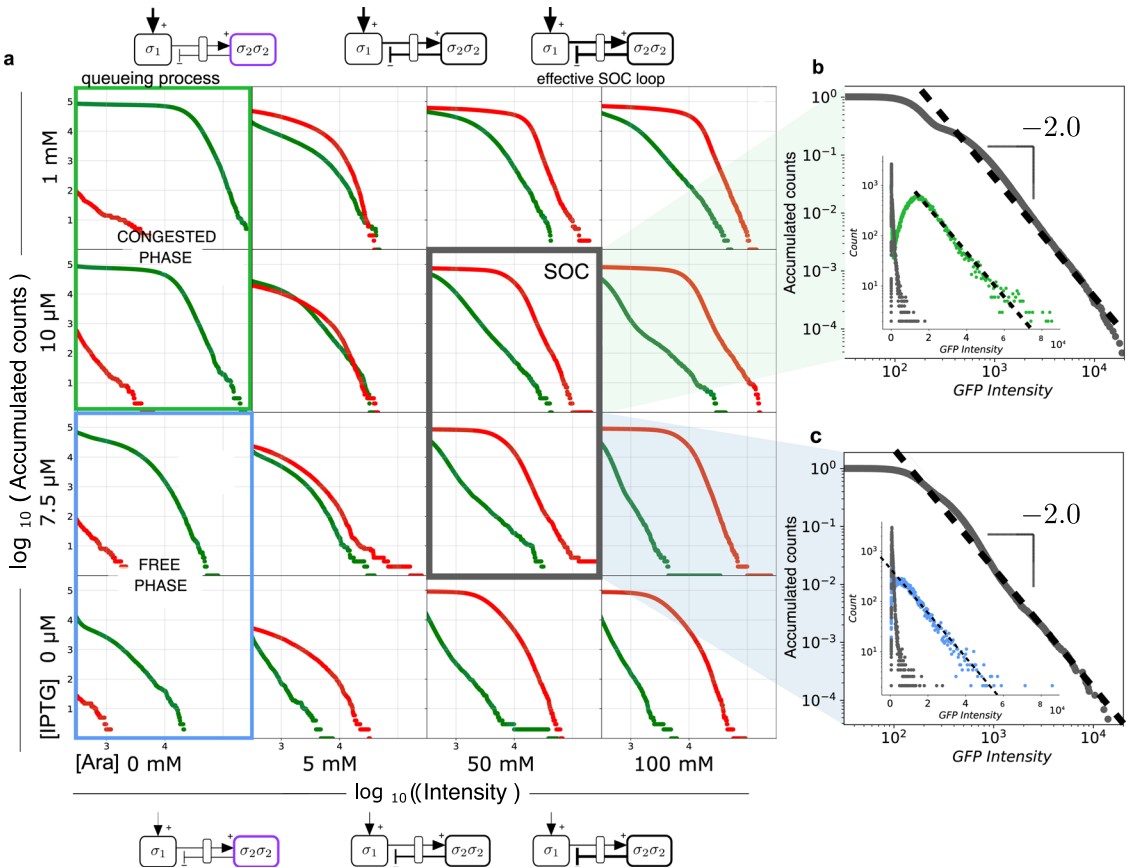

**Fig. 4 Statistics and power-law distributions for SOC from the experiments. a** Cumulative (non-normalized) distributions $P_>(\sigma_{1,2})$ of GFP and RFP fluorescence levels, here plotted using green and red lines, respectively, for the same set of conditions shown in Fig. 3d. The candidate combinations leading to the SOC state (grey panels) are characterized by a broad range of GFP expression revealed by the tail associated with large bursts. In (**b**) and (**c**) two cumulative histograms are shown for (10 $\mu$M IPTG, 50 mM arabinose (Ara)) and (7.5 $\mu$M IPTG, 50 mM Ara), respectively. Both distributions are close to a scaling law $P_>(\sigma_1) \sim \sigma_1^{-2}$ thus leading to a scaling exponent $\gamma \sim 3$, consistent with the stochastic simulations. The insets display the comparison of the raw histograms of the congested state (green dots (**b**)) and free-phase (blue dots,(**c**)), respectively and the SOC state (grey dots) corresponding to the same IPTG conditions. Notice that, unlike the larger images, the x-axes of the insets are linear (so that exponential laws appear as straight lines). Cumulative distributions are shown for fluorescence levels above $10^{2.5}$ (Q2, Q3, Q4 from Fig. 3). All cumulative distributions have obtained from a constant population size of $10^5$ cells. Additional distributions with more detailed IPTG-Ara combinations are shown in Section III.D in the SM and source data are provided as a Source Data file.

power-law in the dynamics of the order parameter $\sigma_1$. From the existence of a transition in the queueing process between the two phases described above, we can conjecture that the SOC motif should easily organize our gene network into this critical boundary provided that the control of the dimer concentration allows the loop to work properly. This is precisely what we found, as shown in Fig. 4a, where we display the same set of Arabinose-IPTG combinations shown in Fig. 3d. Here the statistics of expression are shown in Fig. 4a using cumulative distributions. The histograms of the non-induced *E. coli* colony with low Arabinose but tuned using IPTG (left panels) reveals a single-peak shape (a flat part followed by a rapid decay in the cumulative plot). In general, as we increase the levels of both arabinose and IPTG, the distribution of our order parameter becomes fat-tailed once RFP levels become high, and well-defined power laws can be observed for a wide range of parameters, as we would expect from our SOC motif. By contrast with the standard critical points, where marked differences are apparent in the distributions above and below criticality, all the statistical patterns within our domain of parameters where the SOC motif is effective display scaling laws. The two SOC states highlighted in Fig. 3d are shown in Fig. 4b–c (the insets are linear-log plots to highlight the different

behaviour of the two components of the SOC motif). A detailed parameter exploration is provided in Supplementary Fig. 14–19.

## Discussion

Self-organized criticality (SOC) has a seemingly paradoxical nature: it involves steady states that are always on the edge of instability. Are there intracellular processes poised close to critical points? Traffic dynamics in other contexts suggests that optimal flows occur close to criticality along with very broad fluctuations[22,23]. Within cells, theoretical work suggests that enzymatic networks might be poised to criticality when the substrate input rate is balanced by the processing capacity of the enzymatic network[38] and that SOC states might pervade optimal growth[46,47]. Such critical balance would be a source of adaptation. In this paper, we have followed a constructive approach by building a type of network motif implementing the logic of SOC processes on a two-gene network by following the basic design principle of linking order and control parameters[33]. As the activity level ($\sigma_1$) grows due to overloaded proteolytic machinery, the competition for the ClpXP pool also increases the levels of the control component $\sigma_2$ that can dimerize to perform negative feedback on the emission of $\sigma_1$, thus effectively reducing activity.

Using the SOC motif architecture, it was possible to create a separation of time scales driving to highly fluctuating, critical dynamics. This work shows that this class of unstable attractors can be engineered in living cells. Being at the critical state has important consequences linked with optimality and might be relevant for information-processing tasks. Several authors early suggested that biological computation could occur close to phase transitions[19,48] and given the potential effects of a critical motif on other cellular systems performing given tasks, our results could give support to this conjecture at the cellular level. In this context, Hasty and co-workers have shown that proper engineering of the proteolytic machinery can be used to achieve relevant functionalities, including tunable post-translational coupling[49] or in vivo drug delivery based on pulses of bacterial lysis against colorectal tumours[50,51]. Our SOC motif could further enhance some of these applications (wide fluctuations and rapid responses to external signals). An obvious extension of the critical motif could be a multicellular circuit able to trigger population-level avalanches by exploiting the quorum sensing machinery. Similarly, the fat-tailed behaviour could be wired to a diverse range of functionalities, such as search paths with fat-tailed statistics where bursting dynamics have adaptive value[52,53].

Critical states are known to be part of the cognitive equipment of multicellular organisms, from the simple, non-neural placozoans to neural systems and animal collectives[54,55]. The SOC motif might be an efficient way of generating phenotypic diversity in a microbial population and can be relevant to expand the space of synthetic biology computational designs[56] into collective intelligence[57]. A missing point here is the lack of a time dimension that could help confirming our results and further develop a theoretical framework. This can be achieved by constructing a similar SOC motif within a eukaryotic cell, where the time scale of the resulting time series would be smaller than the cell division cycle. Finally, given the analogies between our system and critical traffic in parallel computer networks, an extension of our approach could involve a 3D spatially explicit system and the development of statistical physics models of critical intracellular traffic.

## Methods

**Plasmids construction**. Plasmid construction and DNA manipulations were performed following standard cloning techniques. The *LacI-lva* (BBa_ C0012), *RFP-lva* (BBa_ K1399001) and *GFP-lva* (BBa_ K082003) genes were amplified from the parts registry collection (2016). The forward primers were synthesized to contain the proper promoter and/or RBS sequences: the *pBAD* promoter and the *RBS30* for the *LacI* gene, RBS34 for *RFP* gene and *pLacIQ* promoter with *RBS34* for *GFP* gene. The PCR products pBad-RBS30-Lacil-lva and RBS3-*RFP*-lva were joined together by assembly PCR, and cloned to pBluescript plasmid in the restriction sites EcoRI and XbaI. The PCR product pLacIQ-RBS34-*GFP*-lva was cloned to a Bluescript plasmid by SpeI and PstI. The resulting plasmids were joined together by ScaI and the blund ends of *Eco53kI* and *EcoRV*. The clonings were realized in the pBluescript II SK(+) plasmid backbone (ColE1 high copy number replication origin). The sequence of primers is shown in the supplementary information (see Section III.A and Figure 12 of Supplementary Materials).

**Strains and growth conditions**. Plasmid cloning and evaluation of the circuit behaviour was performed in the *E. coli* XL1-Blue strain. All characterisation experiments were done in lysogeny broth (LB) Lennox media (10 g/L Tryptone 5 g/L Yeast Extract, 5 g/L NaCl) with a final ampicillin concentration of 125 $\mu$g/mL. Single colonies were inoculated in 4 ml and grown at 37 °C with shaking (200 r.p.m.) during 4 h, to reach an approximate OD$_{660}$ of 0.6. One microliter of the culture was re-inoculated in 4ml of fresh media, supplemented with ampicillin, and the corresponding Arabinose and IPTG concentrations. The cultures were grown overnight (10–14 h) at 37 °C with shaking. Once they were at OD$_{660}$ of 0.8–1, were used for fluorescence measures.

**Imaging of single-cell gene expression**. The output of the SOC circuit was analyzed after 10 h of incubation at 37 °C with different combinations of inputs. Samples were diluted in PBS and analyzed using flow cytometry (BD LSRFortessa; lasers:405-488-561-633, detectors: FSC\SSC + 14PTM, Facs Diva software 9). A total of $10^4$ cells were collected from each sample. Specific emission fluorescence channels for GFP (FITC-H) and RFP (PE-H) were measured. A proper gate to subtract the debris

particles was set using forward and side scattering channels (see Supplementary Fig. 13). For the FACS graphics, the GFP and RFP fluorescence of cells inside the gate were plotted in adjacent axes. The cumulative distributions depict all bacteria with a FITC-H expression above $10^{2.5}$. All data were analysed and plotted using FlowJo (v7) software and customized Phyton code. The regression line and slopes of the histograms were calculated and ploted using Numpy 1.20.3, Matplotlib 3.4.2, Scipy 1.6.3, FlowCytometryTools 0.5.1, Pandas 1.2.4, DiffEqJump v6.14.1 and Catalyst v6.12.1 For microscopic images, the cells were harvested at the same time than the cytometry analysis and pictures were collected with an inverted microscope Leica DMI6000 (Leica LASX v3.3 software), using a 40x oil objective. Bright field, red and green fluorescent images were taken and then merged using ImageJ 1.8.0_172.

**Mathematical modelling**. The mathematical model used here is a two-dimensional system of nonlinear ordinary differential equations describing the coupling between the order ($\sigma_1$) and the control ($\sigma_2$) parameters required to obtain criticality:

$$\frac{d\sigma_1}{dt} = f(\sigma_2) - \delta_1\ \sigma_1 - \sigma_1\ \Gamma(\sigma_1, \sigma_2), \tag{1}$$

$$\frac{d\sigma_2}{dt} = \eta_2 - \delta_2\ \sigma_2 - \sigma_2\ \Gamma(\sigma_1, \sigma_2), \tag{2}$$

where the following Hill function response[58] is used:

$$f(\sigma_2) = \frac{\eta_1}{\theta + \mu^2 \sigma_2^2}, \tag{3}$$

for the repression mediated by $\sigma_2\sigma_2$ dimers. The parameter $\mu \in [0, 1]$ weights the effect of IPTG on the strength of the negative control. When $\sigma_2$ is small (the ClpXP system is working far from congestion) we have a production rate $f(\sigma_2 \rightarrow 0) \approx \eta_1/\theta$. The inhibition function has a threshold value $\theta$ representing the concentration $\sigma_2^*$ at which the rate drops to half its maximum value i. e. $f(\sigma_2^*) = \eta_1/2\theta$. For larger values, it rapidly decays to zero. The saturation function, namely

$$\Gamma(\sigma_1, \sigma_2) = \frac{\delta_c C}{K + \sigma_1 + \sigma_2}, \tag{4}$$

introduces the competition of both proteins for the proteolytic machinery. Here, as well, the limit case when no congestion occurs (due to low concentrations of both $\sigma_1$ and $\sigma_2$) gives a constant removal rate proportional to the concentration of ClpXP units, i. e. $\Gamma(0, 0) = \delta_c C/K$. The expression of $\sigma_1$ gives the behaviour of the GFP-lva. The expression of $\sigma_1$ gives the behaviour of the GFP-lva, whereas $\sigma_2$ stands for LacI-lva. Thus $f(\sigma_2)$ is the expression for the response of pLac (the promoter controlling the expression of GFP) to the LacI protein. In this function, $\eta_1$ is defined as the production rate, $\theta$ the promoter sensitivity, and finally, $\mu$ weights how effective is the repression of LacI (effectiveness being altered by IPTG: the more IPTG the lower the $\mu$ value). The production rate of LacI ($\sigma_2$) is controlled by the *pBAD* promoter, which will trigger a heavier production when there is Arabinose in the medium: the more Arabinose, the higher the value of $\eta_2$. Both proteins are diluted and degraded at rates $\delta_1$ and $\delta_2$ respectively. Finally, both proteins are degraded by the ClpXP system, which can be saturated if there are enough proteins to be degraded. For this reason, there is a sigmoid function, with degradation rate $\delta_c$, $C$ standing for ClpXP concentration and sensitivity $K$. Notice that both proteins compete for the degradation machinery, thus inhibiting each other (being added in the denominator). Stochastic simulations of the previous deterministic model were also implemented using the Gillespie method[39] (see Section II.B, SM).

**Reporting summary**. Further information on research design is available in the Nature Research Reporting Summary linked to this article.

## Data availability

All relevant data are available from the authors. Nonetheless, the source data is available in the open source repository OFS https://osf.io/h5cew/?view_only=33c12a4780954fcd bd9f1c0986adfdc6 Source data are provided with this paper.

## Code availability

The Code needed to do the analysis use the functions precoded in the indicated packages. However, software and custom code used are available upon request.

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

## Acknowledgements

The authors thank Jordi Garcia-Ojalvo as well as the members of the Complex Systems Lab for fruitful discussions. Special thanks to Arianna Bruguera for her help in the experiments. RS thanks S. Kauffman, S. Manrubia, Jordi Piñero and the late Per Bak for many discussions on criticality. This work was supported by the Botin Foundation by Banco Santander through its Santander Universities Global Division, the Spanish Ministry of Economy and Competitiveness, grant AEI-PID2019-111680GB-I00/AEI/ 10.13039/501100011033, an AGAUR FI 2018 grant, and the Santa Fe Institute (where the key idea was conceived). J.S. and A.G. have been partially funded by the CERCA Programme of the Generalitat de Catalunya. J.S. acknowledges support from Agencia Estatal de Investigación grant RTI2018-098322-B-I00 and Ramón y Cajal contract RYC-2017-22243. A.G. has been funded by the AGAUR grant 2017-SGR-1049 and by the MINECO-FEDER-UE grants PGC-2018-098676-B-100 and RTI2018-093860-B-C21.

## Author contributions

R.S. conceived the initial idea, supervised the project and wrote the main paper in collaboration with all the authors. N.C., R.S. and B.V. designed the synthetic cellular circuit. R.S., and B.V. built the two-gene mathematical model. A.G., B.V., R.S., and J.S. performed the mathematical calculations. B.V., A.G., J.S., V.M. and J.P. carried out the simulations and numerical analysis. B.V., V.M. and N.C. performed the experiments.

## Competing interests

The authors declare no competing interests.
