## [Peer Review File · Nature Communications]

Reviewers' Comments:

Reviewer #1:

Remarks to the Author:

The authors start from a theoretical model of SOC, and arrive at its engineering within a genetic regulatory network - to my knowledge, this is the first time this has been achieved. In order to test the success of the engineering action, interestingly the authors created a control system (adjustable from the outside) of the intensity of the relationships between the variables involved in the SOC phenomenon. The work does not expand the knowledge of SOC phenomena: the authors however skilfully show how the realization in a living system of a circuit expressing SOC activity can have very interesting consequences, giving the possibility to achieve relevant functionalities in future broader circuits. Because of these reasons the work is of great interest to authors involved in the system biology community and in other broader scientific communities.

In general, the used statistical approach is appropriate, and the level of detail provided is adequate to assure the reproducibility of the work. As the authors themselves highlight, it was not possible to monitor the SOC phenomenon over time, so the various observations of the work concern the phenomenon spatial dimension (different cells): in case of the ergodicity hypothesis holds (a plausible assumption) the SOC aspect should also occur over time. The actual presence of a temporal aspect could have implications in the possible future application of the circuit.

The work is professional and well written, it clearly establishes its main points, the background is well presented and the methods are adequate to the proposed researches. I therefore only have a few questions on a couple of issues, and a few minor comments

The first point arises from observing figures 3 and 4.

Why do the authors limit the evidence of the SOC phenomenon to the area highlighted by them in figures 3d and 4a? The presence in the same rows of the transition in the queuing process between the congested and the free phase (first column) is fascinating, but for what reasons do the authors exclude the same rows in the last column (or for what reasons do the authors exclude in general the component the last column)? Perhaps the SOC phenomenon is always present for high levels of Arabinose? If not, why? If it is found that the SOC phenomenon is actually (almost) always present for high levels of Arabinose (fourth column of figures 3d and 4a - from the figures it is not clear), is this an important fact for the conclusions of the paper?

The second point concerns an observation that can be made by observing the insets in figure 4b. These insets indicate that the concentration levels of GFP fluorescence levels in the SOC zone are lower than in non-SOC cases (this among other things indicates an efficient degradation action of GFP-lva). However, low concentration levels are known to show higher fluctuations than high concentration levels. If possible, the authors should show that the detected variations belong to a SOC phenomenon more than to a homogeneous phenomenon due to small values. Probably stressing (again...) the power-law aspect of distributions of the concentration values is enough, but if there were other reasons would not be bad.

Minor comments:

- The authors write: "The two regimes are separated by a critical point where an optimal balance is reached, along with wide fluctuations in concentrations". In this way they define the type of criticality relevant to the paper (an area in which large fluctuations of a certain order parameter can be observed). It is a vision consistent with the model that the authors want to implement (a model showing SOC-related phenomena). Perhaps it is appropriate to add a comment indicating that other definitions of criticality are possible, and then indicate some review article for those interested in exploring the topic
- The authors write: "Is a SOC mechanism a possible way of generating critical gene expression

dynamics?" The authors have just commented on the fact that the SOC phenomenon (i) is related to criticality and (ii) shows fluctuations at very different scales. The question thus becomes rhetorical, unless the authors intend to suggest a broader idea of criticality than the one just considered. In that case they could expand the text to make this intention clearer - alternatively, they can simply delete the sentence, also making the text more fluid.

- The authors write: "results from the FACS are displayed in Fig.3c, where..."; the correct reference is Fig.3d
- Caption of Fig.2 – the authors write: "In this panel the hot map is plotted on top of the phase space, showing a maximum close to the attractor as well as...". It is not clear which variable they are referring to with the adjective "maximum"
- Caption of Fig.4: for ease of interpretation the authors should point out that - unlike the larger images - the scale of the x-axis of the insets is linear (so that exponential laws appear as straight lines)

The supplementary materials are adequate and well organized, I have no particular comments about it.

Best regards

Reviewer #2:

Remarks to the Author:

The manuscript by Vidiella et al explored theoretically and experimentally the notion of self-organized criticality (SOC) in a simple two-gene circuit. The idea is to use the second gene to produce a transcriptional repression for the first gene, such that both gene products compete for a finite pool of ClpXP proteases. If ClpXP become over-saturated, the amount of repressor proteins σ_2 would grow, thus reducing the synthesis of σ_1 and thus maintaining ClpXP on the edge of being over-saturated. Authors performed deterministic and stochastic simulations of a corresponding system and have shown that in a certain parameter regime, certain features emerge that can be attributed to SOC. In particular, they found that the distribution of the reporter protein exhibits a power-law like fat tail.

Then the authors tested their idea experimentally by building a two-gene circuit with tagged GFP used as a reporter, and LacI (co-expressed with RFP) used as a repressor for GFP. By varying Arabinose and IPTG that regulate expression of LacI-RFP and its repressivity, respectively, they explored the parameter regime where SOC could be expected, and indeed they also say distributions of GFP fluctuations in FACS experiments that appear to have fat tails.

The manuscript is certainly interesting and thought-provoking, however there are a number of issues that need to be clarified before the decision of its acceptability for Nature Communication can be made.

Major concerns

1. My main concern is how justified is the key assertion that this indeed is SOC and not just a simple criticality. The whole idea of SOC is that the system does need tuning and should reach a critical state itself. However, Fig. 2d shows that there is a clear peak of CV near a particular value of $\eta_2=0.01$, and that's where SOC is claimed. Why is that SOC and not a simple criticality similar to that of [26].
2. To continue that point, it would certainly be worthwhile to compare carefully and contrast the purported SOC in the 2-gene system with simple criticality, when feedback is disabled. Authors appear to have done that by imposing the condition $\mu=0$, however it is not a fair comparison. One should choose parameter values in a control simulation so the expression level of GFP in both

cases is the same, and then compare the statistics of the GFP expression. And repeat that for a range of η_1 , to demonstrate that indeed the system is at a critical state independently of η_1 (in a certain range, at least).

3. SOC regime in Fig. 3d is not explained clearly. It shows correlated fluctuations of GFP and RFP similar to [26]. Why is it an evidence of SOC and not a simple criticality. The same control as above should have been performed in experiments as well.
4. At high levels of ARA the amplitude of GFP seems to get lower (Fig. 3d)
5. Mathematical model could've been used to predict the 2d FACS distributions shown in Fig. 3d, however it was not attempted. Why?
6. Authors refer to Ref. [35] in several places, but they never explicitly state that this paper proposed that SOC can be observed in an adaptive queueing system with feedback. It is a different kind of feedback (affecting the number of ClpXP molecules rather than synthesis rate of the reporter protein itself), but still, the idea is quite similar, and it should be acknowledged.

Minor points

1. In Fig. 2d axis label is obscured.
2. In Fig.2 caption, what is δ ? The model has two deltas, δ_1 and δ_2 . Are they the same? Related: Why the dynamics becomes slow-fast near $\eta_2=0.01$ as mentioned in the text and in the caption to Fig.2? .
3. In Fig. 2, what parameters panel (g) corresponds to? The solid line leading to it seems to originate in a circle at $\eta_2=0.01$ and $CV=0.1$ on panel (d), what is this point?
4. P.6 "this nonlinear flows" should be fixed.
5. p.8 Figs. 3c should be 3d.
6. Reactions (S.6), (S.19), (S.22) should be written as $G+C \rightarrow C$ etc., since ClpXP is not degraded. Or just $G \rightarrow 0$, where C is implicit in the reaction rate.

Responses to the Referees for MS NCOMMS-20-47683

We want to thank the Editor for handling our MS and the two Referees for providing us with feedback. Find please below the responses to the Referees' comments and a detailed explanation of all the changes introduced in the revision. For clarity, our responses below are written in blue colour. As requested by *Nature Communications*, we have highlighted the changes in the manuscript text file as well as in the Supplementary Material (removed and added text appear in red and blue, respectively).

REVIEWER COMMENTS

Reviewer #1 (Remarks to the Author):

The authors start from a theoretical model of SOC, and arrive at its engineering within a genetic regulatory network - to my knowledge, this is the first time this has been achieved. In order to test the success of the engineering action, interestingly the authors created a control system (adjustable from the outside) of the intensity of the relationships between the variables involved in the SOC phenomenon. The work does not expand the knowledge of SOC phenomena: the authors however skilfully show how the realization in a living system of a circuit expressing SOC activity can have very interesting consequences, giving the possibility to achieve relevant functionalities in future broader circuits. Because of these reasons the work is of great interest to authors involved in the system biology community and in other broader scientific communities.

In general, the used statistical approach is appropriate, and the level of detail provided is adequate to assure the reproducibility of the work. As the authors themselves highlight, it was not possible to monitor the SOC phenomenon over time, so the various observations of the work concern the phenomenon spatial dimension (different cells): in case of the ergodicity hypothesis holds (a plausible assumption) the SOC aspect should also occur over time. The actual presence of a temporal aspect could have implications in the possible future application of the circuit.

The work is professional and well written, it clearly establishes its main points, the background is well presented and the methods are adequate to the proposed researches. I therefore only have a few questions on a couple of issues, and a few minor comments

We thank the Referee for her/his positive assessment of our work. Below we provide our responses to the requested queries concerning the definition of SOC (and potential controversies) as well as our interpretation of experimental data as SOC or non-SOC, along with the responses to minor comments.

The first point arises from observing figures 3 and 4.

Why do the authors limit the evidence of the SOC phenomenon to the area highlighted by them in figures 3d and 4a? The presence in the same rows of the transition in the queuing process between the congested and the free phase (first column) is fascinating, but for what

reasons do the authors exclude the same rows in the last column (or for what reasons do the authors exclude in general the component the last column)? Perhaps the SOC phenomenon is always present for high levels of Arabinose? If not, why? If it is found that the SOC phenomenon is actually (almost) always present for high levels of Arabinose (fourth column of figures 3d and 4a - from the figures it is not clear), is this an important fact for the conclusions of the paper?

From the experimental data, it is clear that, once a given level of Arabinose concentration is reached, there is a shift in the RFP expression levels that provide a qualitative condition for the SOC state to emerge. That is indicated in figure 3d by means of the larger gray window where the SOC motif effectively controls the dynamical state and allows broad fluctuations to occur. It is important to notice that, due to the cost and time limitations of the experimental analyses, the x-axis includes discrete values of Arabinose concentrations. In this context, despite that [Ara] in the last column is twice the value of the third column, we keep observing basically the same broad distributions consistent with the expected SOC state. We were indeed very conservative as we noticed a bit more variable distributions, but there is no reason to think that the phenomenon is different. We have changed this accordingly to expand the space of SOC states in figures 3d and 4a.

One specific comment concerning this point has been also added at the end of the results section, which now reads: "In general, as we increase the levels of both arabinose and IPTG, the distribution of our order parameter becomes fat-tailed once RFP levels become high, and well-defined power laws can be observed for a wide range of parameters, as we would expect from our SOC motif. By contrast with the "standard" critical points, where marked differences are apparent in the distributions above and below criticality, all the distributions within our domain of parameters where the SOC motif is effective display fat-tailed distributions."

The second point concerns an observation that can be made by observing the insets in figure 4b.

These insets indicate that the concentration levels of GFP fluorescence levels in the SOC zone are lower than in non- SOC cases (this among other things indicates an efficient degradation action of GFP-Iva). However, low concentration levels are known to show higher fluctuations than high concentration levels. If possible, the authors should show that the detected variations belong to a SOC phenomenon more than to a homogeneous phenomenon due to small values. Probably stressing (again...) the power-law aspect of distributions of the concentration values is enough, but if there were other reasons would not be bad.

The Referee is right in the interpretation of the reduced levels of GFP fluorescence. On the other hand, given the large sets analyzed from the FACS data, we consistently think that the statistics is solid. Different replicas provide a good distribution profile and support this. It is important to notice that in fact, due to the long-tailed character of the distributions, a purely stochastic deviation from a single-scale behavior cannot account for the consistent long-tailed distributions.

Minor comments:

- The authors write: “The two regimes are separated by a critical point where an optimal balance is reached, along with wide fluctuations in concentrations”. In this way they define the type of criticality relevant to the paper (an area in which large fluctuations of a certain order parameter can be observed). It is a vision consistent with the model that the authors want to implement (a model showing SOC-related phenomena). Perhaps it is appropriate to add a comment indicating that other definitions of criticality are possible, and then indicate some review article for those interested in exploring the topic

The Referee is right in pointing to the controversial definition of criticality. As the Referee knows, this is a controversial issue, particularly when dealing with non-conservative systems (as discussed by Bonachela and Muñoz, 2009). This is particularly interesting for neural systems but in general has made necessary to reconsider what is the right term to be used. Since we do not have much space for a serious discussion on this point, we have followed the Referee’s advice and modified the last paragraph of the introduction, mentioning the potential constraints required to have a proper SOC state and added some extra references. These references exemplify the need to adjusting some microscopic features (such as the shape of rice grains in a pile of rice), the effects of increasing rates of driving (and the concept of self-organized instability) as well as a 2015 revision of the state of the art and ongoing controversies.

- The authors write: “Is a SOC mechanism a possible way of generating critical gene expression dynamics?” The authors have just commented on the fact that the SOC phenomenon (i) is related to criticality and (ii) shows fluctuations at very different scales. The question thus becomes rhetorical, unless the authors intend to suggest a broader idea of criticality than the one just considered. In that case they could expand the text to make this intention clearer - alternatively, they can simply delete the sentence, also making the text more fluid.

Indeed, that was superfluous. Deleted

- The authors write: “results from the FACS are displayed in Fig.3c, where...”; the correct reference is Fig.3d

Corrected

- Caption of Fig.2 – the authors write: “In this panel the hot map is plotted on top of the phase space, showing a maximum close to the attractor as well as...”. It is not clear which variable they are referring to with the adjective "maximum"

Yes, the use of maximum here was misleading. First, we have specified in the previous sentence that maximum concerns to the CV. Then, we have substituted the sentence: “*In this panel the hot map is plotted on top of the phase space, showing a maximum close to the attractor as well as..*” by: “*In this panel the hot map is plotted overlapped to the phase space, showing a larger density close to the deterministic attractor as well as...*”. Here, the word *maximum* referred to maximum density values of the hot map. We believe now is clear.

- Caption of Fig.4: for ease of interpretation the authors should point out that - unlike the larger images - the scale of the x-axis of the insets is linear (so that exponential laws appear as straight lines)

Completely agree. We have added this text to the caption.

The supplementary materials are adequate and well organized, I have no particular comments about it. Best regards

Thank you so much.

Reviewer #2 (Remarks to the Author):

The manuscript by Vidiella et al explored theoretically and experimentally the notion of self-organized criticality (SOC) in a simple two-gene circuit. The idea is to use the second gene to produce a transcriptional repression for the first gene, such that both gene products compete for a finite pool of ClpXP proteases. If ClpXP become over-saturated, the amount of repressor proteins σ_2 would grow, thus reducing the synthesis of σ_1 and thus maintaining ClpXP on the edge of being over-saturated. Authors performed deterministic and stochastic simulations of a corresponding system and have shown that in a certain parameter regime, certain features emerge that can be attributed to SOC. In particular, they found that the distribution of the reporter protein exhibits a power-law like fat tail. Then the authors tested their idea experimentally by building a two-gene circuit with tagged GFP used as a reporter, and LacI (co-expressed with RFP) used as a repressor for GFP. By varying Arabinose and IPTG that regulate expression of LacI-RFP and its repressivity, respectively, they explored the parameter regime where SOC could be expected, and indeed they also say distributions of GFP fluctuations in FACS experiments that appear to have fat tails. The manuscript is certainly interesting and thought-provoking, however there are a number of issues that need to be clarified before the decision of its acceptability for Nature Communication can be made.

Major concerns

1. My main concern is how justified is the key assertion that this indeed is SOC and not just a simple criticality. The whole idea of SOC is that the system does need tuning and should reach a critical state itself. However, Fig. 2d shows that there is a clear peak of CV near a particular value of $\eta_2=0.01$, and that's where SOC is claimed. Why is that SOC and not a simple criticality similar to that of [26].

Indeed, this is a major conceptual point that we expected would be controversial. As stated by the Referee (and by ourselves in the paper) SOC is, by definition, a dynamical state where the system poises itself (with no further driving) at the critical state. The definition itself has not been free from dispute and controversy, as we have pointed to Referee #1's comments concerning the proper definition and its candidate alternatives. In this context, we have mentioned the point that some of the "successful" examples of SOC dynamics have revealed the need of necessarily adjusting some microscopic features, as we mention (after Referee #2's comments) at the end of the introduction (with additional references).

More importantly, we aimed at constructing a motif (our *SOC motif*) that could naturally include the feedback between order and control parameters that seem to be a pre-condition for SOC states and in that sense we were not just tuning one rate to move from the free- to the congested phase, as shown in (Cookson et al., 2011). We have added a final sentence at the end of the introduction:

“Given the consensus that the presence of this feedback loop is a pre-condition for SOC, the approach taken here requires locating the SOC motif in the right parameter space (not only a given point) where the scale-free behavior will be the robust outcome.”

that we hope can settle the distinction. Finally, following the comments of Referee #2, we have also highlighted the fact that, once the threshold levels of [Ara] have been achieved, fat-tailed distributions are typical (not only close to the standard criticality transition point).

To further show the emergence of SOC in our system, we have monitored how the "control parameter" ($\hat{\eta}_1$) behaves under different identified parameter values (those showing the peak in the CV and those without the peak). Specifically, we have plotted the time evolution of this control parameter in 3 scenarios: (i) parameters combination located at the peak of the green data in Fig. 2d showing SOC properties; (ii) non-regulated scenario with $\mu=0$; (iii) regulated system far from the CV peak with $\mu = 1.5$ and $\eta_2 = 0.001$. Case (i) shows that this "dynamical" control parameter moves close to the transition boundary (Fig. 2g and Fig. S9). Despite this parameter is monitored using the corresponding stochastic trajectory for each scenario (and is thus fluctuating), here the parameter is clearly poised close to the boundary. This has been shown with the probability of the dynamical parameter to reach values of $\hat{\eta}_1$. As a difference, for scenarios (ii) and (iii) it is clearly shown that the control parameter remains at the congested phase (Fig. S10). These analyses have been commented at the end of Section II.A and Section II.A.3. in the Supplementary Material. We think that these analyses can clarify whether SOC (understood as the capacity of a system to tune itself close to the transition boundary) is occurring under the identified parameter values in our work.

2. To continue that point, it would certainly be worthwhile to compare carefully and contrast the purported SOC in the 2-gene system with simple criticality, when feedback is disabled. Authors appear to have done that by imposing the condition $\mu=0$, however it is not a fair comparison. One should choose parameter values in a control simulation so the expression level of GFP in both cases is the same, and then compare the statistics of the GFP expression. And repeat that for a range of η_1 , to demonstrate that indeed the system is at a critical state independently of η_1 (in a certain range, at least).

We understand the point raised by the Referee but we believe that we would be doing an unfair comparison if we tune the parameters to make GFP levels comparable. This would not properly match the experimental settings nor the fact that this is a dissipative system and that changes in amplitude should be expected as parameters are tuned (but see also below).

3. SOC regime in Fig. 3d is not explained clearly. It shows correlated fluctuations of GFP and RFP similar to [26]. Why is it an evidence of SOC and not a simple criticality. The same control as above should have been performed in experiments as well.

Indeed these are good points that require some extra effort that we have tried to make more clear in the revised paper. We think that with the new analyses discussed above, the evidence of SOC is now clearer (Fig. 2g, Figs. S9-S10). The finding of the control parameter being tuned close to the transition point provides a fingerprint for the SOC behaviour. We would like to emphasize that similar analyses looking for the tunability of control parameters under experimental conditions might be extremely difficult, and performing a whole new set of experiments would take a considerable effort in terms of time and resources while, we suspect, would not provide further insights. Moreover, as we pointed to Referee 1, we must notice that there is a whole domain of the explored parameter space where the scaling laws are robust. This is specifically mentioned at the end of the Results section in the revised version: *"In general, as we increase the levels of both arabinose and IPTG, the distribution of our order parameter becomes fat-tailed once RFP levels become high, and well-defined power laws can be observed for a wide range of parameters, as we would expect from our SOC motif. By contrast with the "standard" critical points, where marked differences are apparent in the distributions above and below criticality, all the distributions within our domain of parameters where the SOC motif is effective display fat-tailed distributions."* If the results were somehow equivalent to a standard transition, we should only see scaling behavior close to the standard critical point. The widespread observation of fat-tailed distributions seems to support our SOC picture.

4. At high levels of ARA the amplitude of GFP seems to get lower (Fig. 3d)

Indeed, this is an interesting observation. When there is more arabinose, the measured cells have lower levels of GFP expression. In principle, this was an expected result because higher levels of Arabinose promote more production of LacI due to pBAD promoter activation. Moreover, in the genetic construct, LacI represses the GFP production, because pLac is the promoter before GFP. However, the interaction with the ClpXP degradation complex makes the results not as would be expected without the Iva tag.

In the dotplot (histogram 2D) of fig 3d, we can understand the amplitude in terms of the intensity of the GFP signal, that is, the percentage of bacteria that have enough intensity of green to appear in the upper quarters (Q2+Q4). This becomes clearer if we represent the data of 50mM ARA (in blue) and 100mM ARA (in red) of these quadrants as a non-cumulative histogram:

In these plots, it is clearly shown that higher Arabinose concentration is associated to lower GFP intensity. This happens because, for high Arabinose concentrations, ClpXP is completely saturated (or almost saturated for low IPTG concentrations). Then, protein dynamics is governed by the promoter's regulation. Otherwise, ClpXP degrades LacI and there is less repression. At the same time, for lower Arabinose levels, LacI and RFP indirectly increase GFP by reducing its degradation via ClpXP.

The observation of the Referee is also seen in the histograms in figure 4 (green lines, GFP). There, the cumulative distributions for all bacteria with fluorescence signal above $10^{2.5}$ is plotted (Q2,Q4). Moreover, the cumulative distribution is obtained from the complete population measures (from 10^5 cells). With constant number of cells, the results consistently show that for high Arabinose concentrations, bacteria with high GFP expression

are very rare.

We have noticed that we stated that number of cells are constant in the Material and Methods section but not in the caption of the figure. We have now corrected that.

5. Mathematical model could've been used to predict the 2d FACS distributions shown in Fig. 3d, however it was not attempted. Why?

We have not presented this because the plots (1) do not reveal extra interesting information and -more importantly- (2) they lack the complex structure displayed by the actual FACS data sets. As it happens with other experimental SOC systems, using a very simple model can provide the required insight and support for the criticality hypothesis (as provided by the tendency to the critical state and power law distributions) but seldom recover the detailed patterns. This is the case, for example, of the classical sandpile or earthquake SOC models, where the spatial organization of avalanches on the pile or the spatial self-similarity of quakes are ignored and replaced by a two-dimensional toy model, which will tell nothing (even close) to that kind of information. We are aware of that and decided to go for the simplest model able to give the needed insight to create the motif and account for the key experimental results, together with the aim of better understanding (qualitatively and quantitatively when possible) the impact of key parameters on dynamics and system properties close to the transition. We are currently exploring an extended version of the model where some features of the molecular machinery (including synthesis-degradation and time scales) might allow to extend our comparison beyond the simpler (but we hope that clever) model described here.

6. Authors refer to Ref. [35] in several places, but they never explicitly state that this paper proposed that SOC can be observed in an adaptive queueing system with feedback. It is a different kind of feedback (affecting the number of ClpXP molecules rather than synthesis rate of the reporter protein itself), but still, the idea is quite similar, and it should be acknowledged.

We thank the Referee for pointing us to this. Indeed, we agree with the Referee that the results by Steiner *et al* (now reference [38]) need to be properly highlighted. After the citation at the end of the first paragraph of Section II.A we added an explicit (and fair) acknowledgement:

“This is actually an instance of self-organized criticality taking place by means of an adaptive queueing system with feedback affecting the number of ClpXP molecules.”

Minor points

1. In Fig. 2d axis label is obscured.
Corrected

2. In Fig.2 caption, what is δ ? The model has two deltas, δ_1 and δ_2 . Are they the same? Related: Why the dynamics becomes slow-fast near $\eta_2=0.01$ as mentioned in the text and in the caption to Fig.2?

This δ should be $\delta_{1,2}$. We have arranged it. We have also changed δ_{Clp} by δ_c .

3. In Fig. 2, what parameters panel (g) corresponds to? The solid line leading to it seems to originate in a circle at $\eta_2=0.01$ and $CV=0.1$ on panel (d), what is this point?

The parameters for panel (g) correspond to the lower black circle, with $\eta_2 = 0.01$ and $\mu = 0$. We have added the parameter values to the caption to clarify this point.

4. P.6 “this nonlinear flows” should be fixed.

Corrected

5. p.8 Figs. 3c should be 3d.

Corrected

6. Reactions (S.6), (S.19), (S.22) should be written as $G+C \rightarrow C$ etc., since ClpXP is not degraded. Or just $G \rightarrow 0$, where C is implicit in the reaction rate.

Corrected.

Reviewers' Comments:

Reviewer #1:

Remarks to the Author:

The authors made considerable efforts to argue that we are indeed dealing with a SOC phenomenon. The fact that the area in which the presence of fat-tailed distributions is observable is relatively large - as the authors point out - is a strong clue towards this conclusion.

Given the data collected, I think that the larger gray window of Figures 3 and 4 of the last version of the paper is however not justified (the boundaries of the zone in which the SOC loop is effective are not known). It is therefore more correct not to draw such boundaries and to remember in the text and in the captions that (i) the distributions in the SOC zone are well delineated, while (ii) outside it they are less so, even if the SOC loop seems to be effective anyway, at least within the limits of the explored parameter space.

Once this last correction is made, I think that the authors have satisfactorily answered the points raised by the reviewers, and that the paper can be published.

Reviewer #2:

Remarks to the Author:

I appreciate the efforts the authors made to improve their presentation. Some of the problems have been fixed. I also appreciate that to perform experiments that I requested in my original report is a serious undertaking, and I leave it to the editor to decide if it is okay not to perform them for this publication.

However, I am still not quite satisfied with the justification of the fact that SOC was indeed observed even in numerical simulations where it is much easier to perform corresponding analyses. As I said in my original report, simply setting $\mu=0$ in the model under consideration immediately takes it far away from criticality, hence they observe Gaussian statistics of fluctuations without fat tails there. This is not a convincing evidence. What I requested to do is to demonstrate that the system stays near criticality. Authors refer to Fig. 2g and S9a where a peak in the distribution of $\hat{\eta}_1$ near 0.02 was observed, but if I understood correctly (it was not clearly articulated), this distribution corresponded to a single set of parameter values. The same simulation should have been done for several different values of η_1 , to show that independent of that, the distribution of $\hat{\eta}_1$ still peaks at the transition point 0.02. In principle, the same should be true if they vary other parameters of the system in a certain range, such as η_2 . As I mentioned in the original report, a fair comparison of cases with and without SOC would be to poise the unregulated system ($\mu=0$) at the criticality by tuning η_1 and/or η_2 , and then vary one of more parameters around this critical point to show that in the unregulated case the system quickly loses criticality, while in the regulated case it does not. I think it is a pretty straightforward calculation that should be performed, as it would demonstrate the main point of the paper directly. I still do not understand why SOC presumably occurs only near a single value of $\eta_2=0.01$ and not in a range, so its variations can be compensated by the feedback loop. Can it be clarified?

I should add that new Figs. S9b, S10 were not clear to me. What does a single black dot mean in Fig. S10a? A set of dots that increase toward $\eta_1=0,01$ and then there are no more dots? Does the distribution suddenly drop to zero at higher $\hat{\eta}_1$? They don't not look like probability distributions to me, unlike the distribution in Fig. S9a. What are the color-coded vertical bars for different values of $\hat{\eta}_1$ with the colors that correspond to different values of $P(\hat{\eta}_1)$ in Figs. S9b and S10b,d? Why for example, the dot in Fig. S10a corresponds to $P(\hat{\eta}_1)=100\%$ and the yellow color in Fig. S10b corresponds to $P(\hat{\eta}_1)=0.01$? Was it an attempt to plot a 2D-distribution of $\hat{\eta}_1$ and σ_1 ? It is not clear from the figure

caption at all. By the way, the color bars there show $P(\eta_1)$ without a hat, which I believe is a misleading typo, since η_1 is a fixed parameter.

I also disagree that Fig. 2g, Figs. S9-S10 clearly demonstrate that the system is in SOC regime (response to my comment 3). They only demonstrate that the system presumably self-tunes to the value of effective promoter efficiency near the critical point. Basically, this is a demonstration of the negative feedback control operation. It would be also important to show that the system displays critical properties (power-law statistics of protein fluctuations) for a range of control parameter values (η_1 or η_2), again, unlike unregulated case, where the power law only observes at one value of the control parameter.

Responses to the Referees for MS NCOMMS-20-47683A

We want to thank again the Editor for handling our MS and the two Referees for providing us with further feedback. Find please below the responses to the Referees' comments and a detailed explanation of all the changes introduced in the new revision. For clarity, our responses below are written in blue color. As requested by *Nature Communications*, we have highlighted the changes in the manuscript text file as well as in the Supplementary Material (removed and added text appear in red and blue, respectively).

REVIEWER COMMENTS

Reviewer #1 (Remarks to the Author):

The authors made considerable efforts to argue that we are indeed dealing with a SOC phenomenon. The fact that the area in which the presence of fat-tailed distributions is observable is relatively large - as the authors point out - is a strong clue towards this conclusion.

Given the data collected, I think that the larger gray window of Figures 3 and 4 of the last version of the paper is however not justified (the boundaries of the zone in which the SOC loop is effective are not known). It is therefore more correct not to draw such boundaries and to remember in the text and in the captions that (i) the distributions in the SOC zone are well delineated, while (ii) outside it they are less so, even if the SOC loop seems to be effective anyway, at least within the limits of the explored parameter space.

We fully agree with the Referee, we do not know exactly the boundaries of the effective SOC loop despite in the experimental parameter space they can be identified. We have removed the larger gray boundary and we have stated in the text (Paragraph 8 in Section II.B) as well as in the caption of these figures the two comments pointed by the Referee.

Once this last correction is made, I think that the authors have satisfactorily answered the points raised by the reviewers, and that the paper can be published.

Reviewer #2 (Remarks to the Author):

I appreciate the efforts the authors made to improve their presentation. Some of the problems have been fixed. I also appreciate that to perform experiments that I requested in my original report is a serious undertaking, and I leave it to the editor to decide if it is okay not to perform them for this publication.

We want to thank the Referee again for his/her comments and feedback. We want to apologize if we did not fully address the Referee's concerns in the previous revision. We have thoroughly addressed all the comments below, point by point, to clarify all the doubts and to perform all the new analyses proposed by the Referee (see below). After doing so we sincerely believe that the paper is much clearer and that the results are more consistent.

However, I am still not quite satisfied with the justification of the fact that SOC was indeed observed even in numerical simulations where it is much easier to perform corresponding analyses. As I said in my original report, simply setting $\mu=0$ in the model under consideration immediately takes it far away from criticality, hence they observe Gaussian statistics of fluctuations without fat tails there. This is not a convincing evidence.

What I requested to do is to demonstrate that the system stays near criticality. Authors refer to Fig. 2g and S9a where a peak in the distribution of $\hat{\eta}_1$ near 0.02 was observed, but if I understood correctly (it was not clearly articulated), this distribution corresponded to a single set of parameter values. The same simulation should have been done for several different values of η_1 , to show that independent of that, the distribution of $\hat{\eta}_1$ still peaks at the transition point 0.02. In principle, the same should be true if they vary other parameters of the system in a certain range, such as η_2 .

The Referee is right. This distribution corresponds to a single set of parameter values, given by: $\eta_1 = 10^{-2}$, $\delta_{1,2} = 5 \times 10^{-2}$, $\delta_c C = 10^{-2}$, $K = \theta = 10^{-3}$. In the new revision we have clarified this point. In the first revision we added other parameter combinations to show some other regions of the parameter space without critical behavior (i.e. without SOC): Fig. S10a (now Fig. S12a) for the non-regulated system, with $\mu = 0$; and Fig. S10b (now Fig. S12b) for a regulated system not able to self-tune to criticality. We also agree that for some other parameter values (e.g., of both η_1 and η_2) the system should be able to tune itself close to the transition point. This will provide more strength to our analyses to evince SOC. Following the suggestion of the Referee, we have performed simulations tracking the dynamical parameter $\hat{\eta}_1$ for different values of η_1 and η_2 , also computing the distributions to show the robustness of the power-law behavior together with the bursting dynamics shown by the time series of σ_1 .

We have placed these analyses in two new supplementary figures right after Fig. S9, since these new analyses complement it. Hence, in the new figure Fig. S10a :

we show our results for nine different values of η_1 while keeping $\eta_2 = 0.01$. These values have been explored over a parameter range that spans more than one order of magnitude. Interestingly, for all chosen η_1 values we have observed that SOC is maintained in a robust way, both in terms of the spiky behavior in the time series as well as in their associated power-laws in the cumulative distributions of σ_1 . As expected from a generic mechanism that pushes the system close to criticality, both the range of values and the exponents for the power-law domain remain the same.

On the other hand, the variation of η_2 within the same range of values gives rise to different behaviors. This is consistent with the previous analyses displayed in Fig. 2 (main manuscript). From the point of view of the comparison between the theoretical/simulation results and our experiments (recall that η_2 is associated to the production rate of pBAD promoter), the results observed by tuning η_2 are -as expected- consistent with changes in the distributions when changing [Ara] in the experiments (Fig. 4 in the main text). The SOC behavior in the simulations and the experiments is found at intermediate values of η_2 and [ARA], respectively.

These additional results are commented in the revised version of the SM, and are also shortly referred in the main text (last paragraph in Section IIA).

As I mentioned in the original report, a fair comparison of cases with and without SOC would be to poise the unregulated system ($\mu=0$) at the criticality by tuning η_1 and/or η_2 , and then vary one of more parameters around this critical point to show that in the unregulated case the system quickly loses criticality, while in the regulated case it does not. I think it is a pretty straightforward calculation that should be performed, as it would demonstrate the main point of the paper directly. I still do not understand why SOC presumably occurs only near a single value of $\eta_2 = 0.01$ and not in a range, so its variations can be compensated by the feedback loop. Can it be clarified?

See below: this is answered together with (basically the rest) of other questions below

I should add that new Figs. S9b, S10 were not clear to me. What does a single black dot mean in Fig. S10a? A set of dots that increase toward $\eta_1=0,01$ and then there are no more dots? Does the distribution suddenly drop to zero at higher $\hat{\eta}_1$? They don't not look like probability distributions to me, unlike the distribution in Fig. S9a. What are the color-coded vertical bars for different values of $\hat{\eta}_1$ with the colors that correspond to different values of $P(\hat{\eta}_1)$ in Figs. S9b and S10b,d? Why for example, the dot in Fig. S10a corresponds to $P(\hat{\eta}_1) = 100\%$ and the yellow color in Fig. S10b corresponds to $P(\hat{\eta}_1)=0.01$? Was it an attempt to plot a 2D-distribution of $\hat{\eta}_1$ and σ_1 ? It is not clear from the figure caption at all. By the way, the color bars there show $P(\eta_1)$ without a hat, which I believe is a misleading typo, since η_1 is a fixed parameter.

The point made by the referee, namely the need of carefully looking at the behavior of the unregulated transition, has made a big difference in the final logic and consequences of our work. While exploring the main point raised by the referee, we found something that we did not know. We were familiar, from previous work by some of us (RS) that the latencies (time required for a given packet from delivery to whole processing within a queue) in a critical queueing process followed power laws. We somewhat assumed that -as it occurs in many typical examples of phase transitions- the rest of the statistical behavior would also be scale-free.

We should have reconsidered that while looking at the standard random-walk-like plots of queue time series: even close to the free-congestion transition, time series appear to fluctuated broadly, but never in the bursting pattern that is common to all the examples cited in our paper as a universal feature of SOC. And it turns out -as discussed here in the revised version- that this is indeed not the case. We have carefully commented this, adding new references (the classical works from mathematical models of queueing theory are so many):

- [14] Hesse J., Gross T. 2014. Self-organized criticality as a fundamental property of neural systems. *Front. Neurosci.* 8, 166.
- [15] Mukherjee, G. and Manna, S.S., 2005. Phase transition in a directed traffic flow network. *Phys. Rev. E* 71. 066108.

In the revised SM, we have modified our previous figure S2 to incorporate both the wider view to the transition between the two phases (Fig. S2a) as well as the stationary distributions for different values of η_1 :

The non-power law nature of these unregulated systems (exponential on the free-phase regime and Gaussian in the congested one) changes our narrative in a very positive way, and solves many of the queries raised by the referee. From the statistical side, the SOC state is characterized by the presence of scaling laws that are absent in the activity distributions for the non-regulated system. From the dynamical, time-dependent side, bursting dynamics is also a consequence of the phase separation found at criticality. As we point out in the revised version, it is in fact noticeable that the experimental results for the unregulated domain suggest precisely the presence of single-scale statistics.

What are the consequences? We have carefully tried to introduce this in the revised manuscript, by emphasizing from the beginning the mathematical findings related to Gaussian statistics in queues and as a reminder that the lack of scale-free statistics means that no bursting dynamics will be observed (something that we look for here as a main attributed of SOC dynamics), see our revised introduction:

molecules 'waiting' to be processed will be present (congested phase). The two regimes are separated by a narrow parameter domain (Fig.1b; see also Fig. S2) where an optimal balance is reached, along with broad fluctuations in concentrations [27]. However, they do not follow power laws [28] but exponential-tailed forms $P(s) \sim \exp(-s/s_c)$, see Fig. S2b-f, with s_c rapidly increasing as we approach criticality [15]. Here scaling is found to occur instead in the distribution of latencies, i. e. the time required from the production to the final processing of each particle [24].

The main point is that our mechanism is not only taking advantage of the free-congested transition: it actually transforms its non-critical character into a true, SOC behavior were scaling laws appear:

molecules 'waiting' to be processed will be present (congested phase). The two regimes are separated by a narrow parameter domain (Fig.1b; see also Fig. S2) where an optimal balance is reached, along with broad fluctuations in concentrations [27]. However, they do not follow power laws [28] but exponential-tailed forms $P(s) \sim \exp(-s/s_c)$, see Fig. S2b-f, with s_c rapidly increasing as we approach criticality [15]. Here scaling is found to occur instead in the distribution of latencies, i. e. the time required from the production to the final processing of each particle [24].

And further emphasized at the end of the theoretical section:

S10-S11). However, there are no true phases now: the system bounces back and forth between fluid and congested states as it tends to get close to criticality (see also Section II.A.3, SM). By exploiting the SOC motif, which allows reducing the rate of σ_1 production, the Poissonian dynamics of the original, non-regulated queueing dynamics is transformed into a bursting signal with fat-tailed statistics.

This of course changes our narrative and solves many of the queries raised by the referee. Although we maintain the use of the $\hat{\eta}_1$ dynamical control parameter, we have removed the previous figures discussing its use (Fig. S10 in the first revision; we agree with the referee that were difficult to interpret) and only kept the main one (used in figure 2) while we also make use it as a complement to other measures applied to a wider exploration of parameter space as the Referee suggested, summarized in figures S10 and S11 in the Suppl. Mat.

I also disagree that Fig. 2g, Figs. S9-S10 clearly demonstrate that the system is in SOC regime (response to my comment 3). They only demonstrate that the system presumably self-tunes to the value of effective promoter efficiency near the critical point. Basically, this is a demonstration of the negative feedback control operation.

Again, we agree with the referee that our argument got tangled with the use of these graphics. Given the new scenario presented here, there is no need for these stochastic implementations.

It would be also important to show that the system displays critical properties (power-law statistics of protein fluctuations) for a range of control parameter values (η_1 or η_2), again, unlike unregulated case, where the power law only observes at one value of the control parameter.

As mentioned, we have expanded the exploration of our model (see above) but the main point is that in the unregulated case we do not have power laws.

We thank the reviewer for insisting in clarifying the distinction between our SOC motif implementation and the “classical” transition that we expected to observe for the unregulated set of conditions. It was a much needed exploration and has revealed something unexpected that makes our argument stronger while opens the door for further theoretical exploration. By doing this exploration, not only the quality (and we hope the clarity) of the presentation has improved. Novel conceptual possibilities have emerged.

What seemed obvious was not obvious at all. We have learned some important lessons here and we want to thank the Referee for driving us to this point of view and interesting results.

Reviewers' Comments:

Reviewer #2:

Remarks to the Author:

Authors did a very good job clarifying the points raised by me and other referee and adding illuminating new material in the second round. I think the manuscript is now acceptable for publication in Nature Communications